

# SCShores: A comprehensive shoreline dataset of Spanish sandy beaches from a citizen-science monitoring program

Rita González-Villanueva[1], Jesús Soriano-González[2], Irene Alejo[1], Francisco Criado-Sudau[2], Theocharis Plomaritis[3], Àngels Fernàndez-Mora[2], Javier Benavente[4], Laura Del Río[4], Miguel Ángel Nombela[1], Elena Sánchez-García[2]

[1]Centro de Investigación Mariña, Universidade de Vigo, XM1, 36310 Vigo, España
[2] Balearic Islands Coastal Observing and Forecasting System (SOCIB), Palma de Mallorca, Spain
[3] Department of Applied Physics, Faculty of Marine and Environmental Sciences, University of Cádiz, 11510 Puerto Real, Cádiz, Spain.
[4] Department of Earth Sciences, Faculty of Marine and Environmental Sciences, University of Cádiz, 11510 Puerto Real, Cádiz, Spain.

*Correspondence to*: Rita González-Villanueva (ritagonzalez@uvigo.gal)

**Abstract.** Sandy beaches are ever-changing environments, as they experience constant reshaping due to the external forces of tides, waves, and winds. The shoreline position, which marks the boundary between water and sand, holds great significance in the fields of coastal geomorphology, coastal engineering, and coastal management. It is crucial to understand how beaches evolve over time, but high-resolution shoreline datasets are scarce, and establishing monitoring programs can be costly. To address this, we present a new dataset of shorelines for five Spanish sandy beaches, located in contrasting environments, derived from the CoastSnap citizen-science shoreline monitoring program. The use of citizen science is an increasingly strong current within environmental projects that allows both community awareness and the collection of large amounts of data that are otherwise difficult to obtain. This dataset includes a total of 1721 individual shorelines composed of points every 3 m-spaced points alongshore, accompanied by additional attributes, such as elevation value and acquisition date, allowing for easy comparisons. Our dataset offers a unique perspective on how citizen science can provide reliable datasets that are useful for management and geomorphological studies. The shoreline dataset, along with relevant metadata, is available at https://doi.org/10.5281/zenodo.8056415 (González-Villanueva et al., 2023b).

## 1 Introduction

Low-lying coastal zones are the most intensively used regions on Earth for supporting human population, activity, and industry (Small and Nicholls, 2003). In the European Union (EU) area, 19% of the population lives within 10 km of the coastal fringe (EEA, 2006). Human pressure contributes to the detriment of coastal environmental systems, making it challenging for coastal managers to find a sustainable balance between the fundamental needs of human and natural coastal systems. Moreover, the





latter are highly vulnerable to wave and storm-induced threats. The presence of coastal depositional systems makes those areas more resistant to these hazards by sheltering them from the full impact. The state of these coastal systems is crucial because it provides the first line of defence against flooding of the hinterland (Arkema et al., 2013; Barbier et al., 2011; Vousdoukas et
al., 2020). Considering that sea-level rise and changes in storm tracks, intensities and frequencies are linked to climate change scenarios, the state of coastal systems should be a worldwide concern because the risks in these highly populated areas are likely to intensify (Oppenheimer et al., 2019; Ranasinghe, 2016; Reguero et al., 2019; Theuerkauf et al., 2014).

Currently, coastal monitoring programs and sustainable coastal management are priority topics, not only for Spain coastal managers (DGC, 2008; MITECO, 2020) but also for all EU member states (Farcy et al., 2019). These programs harvest the
knowledge of in-situ surveys and remote sensing observations to better understand change and evolution at different spatiotemporal scales, aiming to improve land-use management and infrastructure planning. Extended knowledge of coastal processes has led to increased concern about the state of beaches as highlighted in the EU strategy on adaptation to climate change (EEA, 2020). Indeed, research that more quickly and comprehensively stresses the fight and adaptation to the unavoidable impacts of climate change are priority objectives of the EU Action. As such, they are clearly expressed in the
Atlantic Action Plan Pillar 4, Goal 6 (European Commission, 2020): Stronger coastal resilience, sharing the best practices on the application of maritime spatial planning to coastal adaptation, resilience, and applicable environmental assessments. Additionally, the Ocean Decade Plan (2021-2030) (https://www.oceandecade.org/), promoted by the United Nations, establishes 10 challenges for collective impact. These challenges include unlocking ocean-based solutions to climate change, increasing community resilience to ocean hazards, expanding the global ocean observing system, and ensuring skills,
knowledge, and technology for all. Therefore, establishing low-cost and accessible monitoring programs for coastal areas is vital to provide the base data needed to understand the resistance and resilience of sandy beaches. Such understanding is crucial for achieving the objectives of coastal management and sustainable planning.

Extensive research has shown that coastal systems evolve over time as changes occur in sediment supply and vegetation cover, and their variation mainly depends on primary drivers, namely waves and winds which in turn show interannual to multi-
decadal variability (Castelle et al., 2022; Davis and FitzGerald, 2004; González-Villanueva et al., 2017, 2023a; Masselink and Huges, 2003; Short, 1999). Despite the broad knowledge of the processes and drivers governing these systems, understanding the sandy beach state requires in-situ measurements of its characteristics and information about beach resistance, recovery, and robustness to offer practical guidance on estimating resilience (Anon, 2022; Kombiadou et al., 2019; Pimm et al., 2019; Masselink and Lazarus, 2019). However, traditional coastal monitoring methods are costly and time-consuming. As a result,
the use of combined high-resolution (local) and moderate-resolution (regional) photogrammetry and remote sensing data for coastal monitoring is gaining traction due to its inherent advantages, and more robust methodologies and accessible tools are being developed for both coastal video imaging systems (Andriolo et al., 2016, 2019; Montes et al., 2018, 2023; Sánchez-García et al., 2017) and satellite imagery (Andriolo et al., 2016; Sánchez-García et al., 2019, 2020; Vos et al., 2019).

Nevertheless, the potential of these methods for coastal monitoring has not yet been fully exploited, and their effective
utilization requires professionals with high levels of expertise.

In addition, citizen science projects have been growing in number and importance over the past decade. These types of
initiatives not only allow obtaining a large number of data that would otherwise be very expensive or impossible to obtain, but
also serve to raise awareness among the general public about some problem, usually of an environmental nature. There are
four common features of citizen science practice: (1) anyone can participate, (2) participants use the same protocol so data can
be combined and be high quality, (3) data can help real scientists come to real conclusions and (4) a wide community of
scientists and volunteers work together and share data to which the public, as well as scientists, have access (Flagg, 2016).

Given the current challenges posed by the limited availability of in-situ observation data for understanding shoreline response,
and the significant advancements in smartphone camera lens technology together with the increasing use of social media, the
CoastSnap international citizen-science initiative emerged in 2017 as a low-cost cutting-edge shoreline mapping approach that
leverages crowdsourced images (Harley et al., 2019). In Spain, the CoastSnap network was established in 2018 at Rodas Beach
(National Park of the Atlantic Islands of Galicia) and has grown moderately since then. The support of a project funded by the
Spanish Foundation for Science and Technology (FECYT) named "Centinelas de la Costa" (FCT-20-15835) facilitated its
expansion since 2021 and the addition of new stations for nationwide dissemination. Currently, there are 22 CoastSnap stations
(CS) along the Spanish Atlantic and Mediterranean coasts completing the network (Figure 1), which is expected to expand in
the forthcoming years. This paper introduces SCShores, a Spain Citizen science Shoreline dataset of geographically corrected
shorelines obtained for five stations of the Spanish CoastSnap network. SCShores includes shoreline positions from sandy
beaches with meso- to micro-tidal regimes, representative of the northwest, southwest, and east coasts of Spain. A total of
1721 individual shorelines composed of points every 3 meters along the shore are delivered, accompanied by additional
attributes such as tide level, timestamp and image source, which allow for easy exploitation. The paper offers a comprehensive
description of the process involved in building the SCShores dataset, along with its limitations and the potential applications
for different end-users. SCShores v.1.0 is designed to be analysed easily using data science tools and geographic information
systems (GIS) software.

## 2 Methods

### 2.1 Study sites and CoastSnap stations settings

The SCShores v.1.0 dataset comprises five Spanish CoastSnap stations that were specifically chosen for this project. Three of
these CS are located in mesotidal beaches, while the remaining two are in microtidal beaches (Fig. 1 and Table 1). These five
stations were selected based on having a larger amount of available images and validation data than the others, with
consideration given to both meso- and micro-tidal beaches. A brief description of each coastline sector and its corresponding
station is provided below.




Mesotidal beaches:

1. *Rodas beach*: Situated on the sheltered eastern margin of the Islas Cíes archipelago at the mouth of the Ría de Vigo
   on the northwest coast of Iberia, this beach forms part of a sand barrier characterized by a well-developed foredune
   that surrounds a shallow saline lagoon. The sandy barrier serves as a natural connection between the two northern
islands within the archipelago and is recognized as a significant ecological feature within the National Park of the
   Atlantic Islands of Galicia. With 1.2 km in length, this beach is classified as a low-energy beach with reflective
   morphology (Costas et al., 2005) and a spring tidal range that can reach up to 4 m. Tourist activity is primarily
   concentrated during the summer season. The CoastSnap station (*cies*), installed in April 2018, is positioned on an
   elevated wooden pathway across the dunes in the northern region, providing a southward view of the beach. (Fig. 1).
Using GPS RTK-GNSS, a total of 10 ground control points (GCPs) and 14 in-situ water lines were measured for
   rectifying the obtained images and validating the mapped shorelines, respectively.

2. *Agrelo beach*: Located on the northwest coast of Spain in the inner part of Ria de Pontevedra (Fig. 1), this coast is
   under a meso-tidal regime, with a spring tidal range of 4.5 m. It is classified as a reflective beach and often presents
   beach cusps and beach steps. The beach is situated in a densely populated area that experiences a significant influx
of tourists during the summer season. It is also backed by human-made infrastructure, including houses and a
   walkway. The CoastSnap station (*agrelo*), installed in January 2019, is positioned on a viewpoint at the southern
   entrance of the beach, providing a northeast beach view (Fig.1). Using GPS RTK-GNSS, a total of 16 GCPs and 11
   in-situ water lines were measured.

3. *Santa María beach*: Located on the southwest coast of Spain, Santa Maria del Mar is an urban pocket-beach, NNW-
SSE oriented, located in the city of Cádiz (Fig. 1). It has a length of 600 m and is strongly controlled by morphological
   contour conditions. It is backed by a high cliff protected by a revetment with a sea wall at its toe. The beach is limited
   at its northern and southern ends by two jetties normal to the shoreline. It is classified as an intermediate-dissipative
   beach with common presence on intertidal sandbars. This is a low energy coast with a mean spring tidal range around
   3 m. The CoastSnap station (*cadiz*), installed in October 2020, is positioned on a viewpoint at the northern entrance
of the beach, providing a southeast beach view (Fig. 1). Using GPS RTK-GNSS, a total of 7 GCPs and 3 in-situ water
   lines were measured.

Microtidal beaches:

S'Amarador and Arenal d'En Tem beaches in Mallorca Island (NW Mediterranean), are characterized by microtidal
conditions with a tidal range of ~ 0.2 m. However, sea levels vary between 0.5 to 1 m due to multiple factors such as
atmospheric and wind-induced surges, inter-annual variations in sea temperature, internal oscillations in the Mediterranean
basin, and the interaction of internal currents in the Western Mediterranean (Haddad et al., 2013; Orfila et al., 2005). During
the tourist season (June to September), both beaches experience high levels of occupancy.



4. *S'Amarador beach*: Located on the Mondragó Natural Park (Fig. 1), S'Amarador is an enclosed sandy beach associated with a gully and constitutes the barrier of a small lagoon. The CoastSnap station (*samarador*) was installed in July 2022; it is positioned on a walkway at the north of the beach, providing a longitudinal southeast view of the shoreline (Fig. 1). Using GPS RTK-GNSS, 4 GCPs and 5 in-situ water lines were measured.

5. *Arenal d'En Tem beach*: Located on the Es Trenc-Salobrar Natural Park, Arenal d'En Tem is part of one of the most emblematic coastal stretches of Mallorca Island (Fig. 1). From a morphodynamic perspective, this beach alternates intermediate and reflective states in winter and summer seasons, respectively. The CoastSnap station (*arenaldentem*) was installed in September 2022; it is positioned on a walkway at the northwest end of the beach (towards seawater), providing a southeast view (Fig. 1). Using GPS RTK-GNSS, 8 GCPs and 2 in-situ water lines were measured.

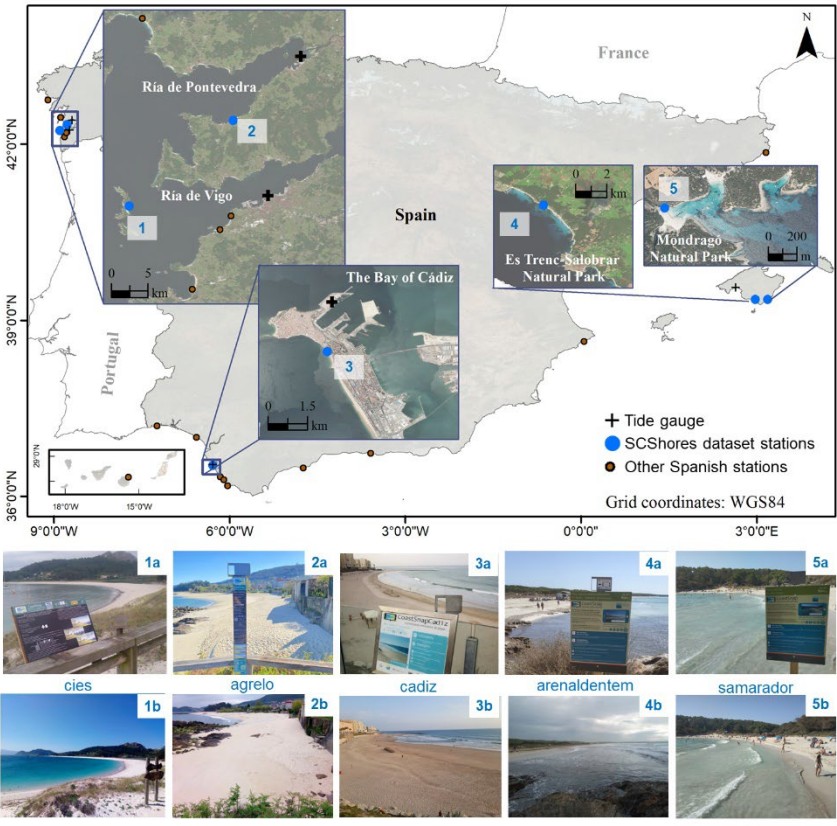

**Figure 1:** Figure 1: Spanish CoastSnap Network including all existing stations until May 2023 (represented by blue and brown dots), along with the tide gauges (depicted as black crosses) used in the SCShores v1.0 dataset, which corresponds to the five stations in the SCShores dataset. (a) Photographs of the SCShores CoastSnap stations, and (b) beach views from each station. Background for the location maps are orthophotos from PNOA sources (CC-BY 4.0 scne.es 2021) and the European boundary layer is downloaded from "http://www.efrainmaps.es. Carlos Efraín Porto Tapiquén. Geografía, SIG y Cartografía Digital. Valencia, España, 2020."





**Table 1:** Beach characteristics and CoastSnap stations (CS) used for the SCShore dataset. The position of the CS is given in World Geodetic System 1984 coordinates (CS_Lon, CS_Lat), and elevation (CS_Z) is referred to the mean sea level in Alicante (NMMA; Spanish vertical datum).

| Beach | Type | Beach face slope | Tidal regime | CS_name | CS_Lon (°) | CS_Lat (°) | CS_Z (m) |
|---|---|---|---|---|---|---|---|
| Agrelo | urban | ~ 0.110 | mesotidal | *agrelo* | -8.772 | 42.331 | 7.154 |
| Rodas | natural | ~ 0.120 | mesotidal | *cies* | -8.900 | 42.226 | 9.633 |
| Santa María | urban | ~ 0.025 | mesotidal | *cadiz* | -6.288 | 36.522 | 16.960 |
| S'Amarador | natural | ~ 0.075 | microtidal | *samarador* | 3.185 | 39.350 | 3.064 |
| Arenal d'En Tem | natural | ~ 0.060 | microtidal | *arenaldentem* | 2.974 | 39.353 | 2.747 |

## 2.2 Dataset generation

The process of generating the dataset involved five main steps: data collection, data curation, data processing, dataset compilation and dataset validation. The dataflow diagram outlining the process for creating SCShores_v1.0 is summarized in Fig. 2.

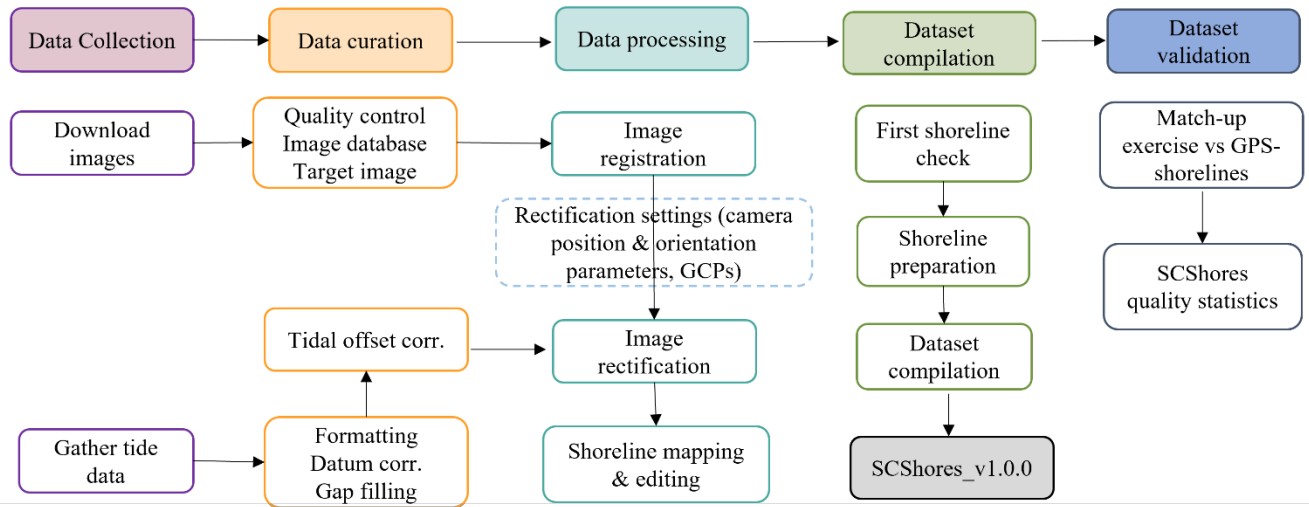

**Figure 2:** Data flow diagram for the generation of SCShores dataset.

### 2.2.1 Data collection

The data collection process adheres to the standards established in the CoastSnap global citizen program, which monitors changing coastlines (Harley et al., 2019; Harley and Kinsela, 2022). At each CS, users are given specific instructions on how to share their images which includes social media (Instagram, Twitter and Facebook), email or web, and smartphone dedicated application (CoastSnap App). A different procedure has been followed for downloading images based on the sharing medium

chosen by the user. As the study beaches have varying tidal regimes, it is necessary to maintain a tidal record to establish the tide level for each photo. This is used in the image rectification step (Fig. 2) and allows the end-users to conduct temporal



analyses under the same sea level conditions. The tidal record was gathered from the closest available tide gauges (Table 2). Due to the absence of local tide gauge at the beach, local tidal offsets were calculated for each beach by computing the difference between the in-situ measurements obtained using GPS RTK-GNSS and the corresponding tide gauge records for

the same time. The validity of the obtained offsets was subsequently established by cross-checking these in-situ tidal elevation measurements against the corresponding tidal elevation value of the timely-closest available CoastSnap image (Fig. 3). The shorelines obtained from the *agrelo*, *cies* and *cadiz* CS, which are mesotidal, may exhibit greater displacement from the reference line due to the temporal variations in tidal levels and the availability of multiple measured shorelines for the same date.


**Table 2:** Tide-gauge locations and characteristics used in SCShores. The tidal offset for each study site was obtained by calculating the difference between the in-situ measurements obtained using GPS RTK-GNSS and the corresponding tide-gauge records for the same time. ΔZ (m) is the difference between each tide gauge zero reference and the NMMA (Spanish vertical datum).

| CS | Tide gauge | Source | Lon (º) | Lat (º) | ΔZ (m) | Local tidal offset (m) |
|---|---|---|---|---|---|---|
| *agrelo* | Marín | Puertos del Estado | -8.690 | 42.410 | -1.802 | 0.40 |
| *cies* | Vigo | Puertos del Estado | -8.730 | 42.240 | -1.772 | 0.30 |
| *cadiz* | Cádiz | Instituto Español de Oceanografía | -6.287 | 36.540 | -1.792 | 0.25 |
| *samarador* | Palma | Puertos del Estado | 2.640 | 39.560 | 0.163 | 0.04 |
| *arenaldentem* | Palma | Puertos del Estado | 2.640 | 39.560 | 0.163 | 0.04 |

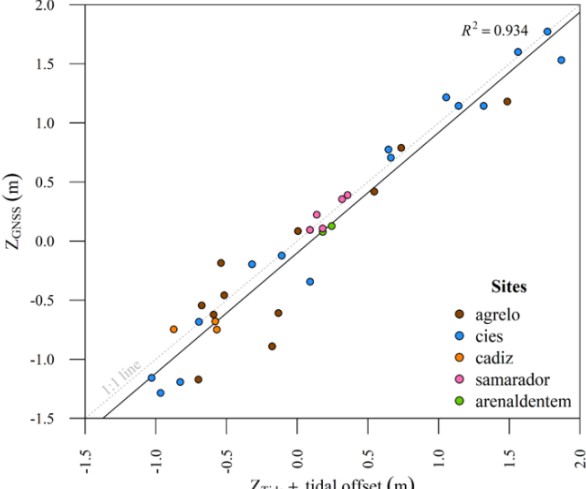


**Figure 3:** Comparison of estimated tidal elevation used as the CoastSnap shoreline elevation ($Z_{Tide}$ + tidal offset) with their corresponding measured shoreline elevation values ($Z_{GNSS}$) for the same dates (refer to coincident days in Fig. 8).

**2.2.2 Data curation**

The data provided by CoastSnap users comes from multiple sources and different models of smartphone. Given that most of the image downloading process is automated, it is necessary to perform a manual verification of the received data to eliminate



duplicate records, incorrect captures, or irrelevant images. Additionally, quality control over the date/time of the snap must be addressed to ensure the correct tide level assignment in photos which lost the original metadata, such as those downloaded from social media (Facebook, Instagram, Twitter). A time stamp quality (TSQ) attribute was set for each image. A TSQ=1

was assigned to images for which the users specified date and time in comments in social media posts and emails, and to images that preserved original metadata or were uploaded to the CoastSnap App which requires the user to confirm the capture date and time. A TSQ = 2 was assigned to images which are expected to be uploaded/sent at the time of capture but were not accompanied by any comment. Consistency between defined datetime and the lighting conditions depicted in the images was also visually checked for each image. Images which were considered unreliable or lacked date or time descriptors were

rejected.

The data gathered from tide gauges need to undergo quality control before they can be used (Fig. 2). This included: (1) homogenizing temporal resolution (to 15 minutes), time zone (transformation to UTC), and vertical datum reference (corrected to the sea level reference, NMMA; Table 2); (2) filling gaps in tidal records using the Ttide routine (Pawlowicz et al., 2002) and linear fitting tide predictions with measured tide; and (3) formatting the data series structure to meet the requirements of

the CoastSnap-toolbox.

### 2.2.3 Data processing

The data processing flow comprises three consecutive procedures (Fig. 2): image registration, image rectification, and shoreline mapping and editing.

The image registration process involves aligning all available images per station with a target image. This process was

performed using Adobe Photoshop software. The registered images were georectified to transform the image from pixel coordinates (u,v) to world metric coordinates (X,Y,Z). To accomplish this transformation, GCPs associated with known pixels of the target image, the position of the station –both measured accurately in the field with a GPS RTK-GNSS–, and the station-view orientation parameters measured with a spirit level are required. This process was performed by using the CoastSnap-toolbox, based on the procedure described by Harley et al. (2019). The dynamic shoreline indicator extracted from these images

corresponds to the instantaneous upper edge of the swash zone that it is highly dependent on the tidal level at the time of photo capture. The toolbox employs the tide elevation value as the Z coordinate or projection value, considered as the sum of the tidal record and the tidal offset corresponding to the date and time of the captured image. This approach accurately locates in the rectified image those features with such Z value, *i.e.*, the shoreline.

CoastSnap-toolbox employs an automatic method for shoreline mapping based on a dynamic threshold applied to the Blue

minus Red band combination, which is intended to maximize the difference between dry and flooded surfaces. The performance of this method can be influenced by various factors, including light conditions, the presence of wet sand, flooded areas, shadows, and wave conditions. To ensure data consistency, we specifically focused on mapping the dynamic shoreline, thus excluding terraces or flooded beach onshore areas (Fig. 4). As it is not always a straightforward interface between sand

and water, all mapped shorelines underwent meticulous examination by an expert and were edited as needed. The shoreline
vertices were manually repositioned to adhere to the dynamic water condition and were interpolated using cubic splines,
assuring that the lines intersected all manually adjusted vertices and achieved an average point spacing of 3 m.

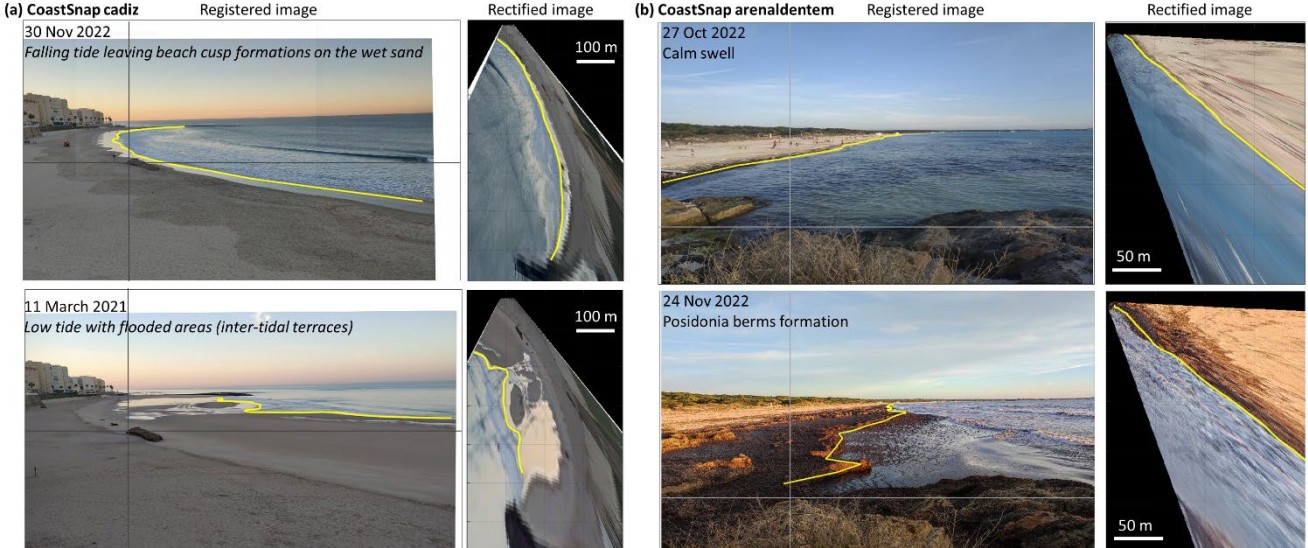

**Figure 4:** Examples of four different scenarios in (a) the "CoastSnap *cadiz*" and (b) the "CoastSnap *arenaldentem*" stations, where the
"dynamic-water" shoreline is mapped (yellow line) in both the registered (oblique image) and rectified images (planview image).

### 2.2.4 Dataset compilation

Before assembling the edited shorelines into a unified dataset, it was essential to identify and exclude shoreline segments
where cumulative errors from image acquisition properties and processing methods could introduce significant inaccuracies.
Since data provided by users are captured from different smartphones, with different camera focal lengths, the resulting images
have different dimensions in terms of width and height (*i.e.,* pixel count). In the best-case scenario, original dimensions are
preserved, such as in the images sent by email with full resolution. However, in most cases, the original pixel size is not
retained due to image compression policies enforced by social media platforms and the CoastSnap App. Platforms like
Instagram, Twitter and the CoastSnap App typically limit image dimensions to a maximum width of 1080, 1200, and 1920
pixels, respectively. Following the registration process, all images are standardized to the resolution of a corresponding target
image (Table 3). However, larger disparities between the target and registered images require further significant interpolations,
resulting in greater deformations and increased uncertainty in the final pixel positions.

Apart from image resolution, the view of the capture (perspective) is also an important parameter to consider for understanding
the error propagation during the resection (i.e., recreating the geometry of a photographic shot) and rectification processes.



The closer to the zenithal plane the image is taken, the lower is the distortion among pixels in the image. In this work, this is related to the elevation of the CS at each site and the length of the shoreline (distance to the camera). Broadly, larger focal lengths and higher elevation led to higher resolution in the georectified image, resulting in smaller long-shore pixel footprint. This is illustrated in Table 3 and Fig. 5, where long-shore pixel footprints are calculated according to Holman and Stanley (2007) formulations across different pixels widths and variable camera elevations –including those that apply to the present

study cases–, demonstrating the relevance of both parameters in reducing the far field pixel size, and thus the plausible resolution of the extracted shoreline.

**Table 3:** Factors that limit the spatial resolution of georectified images. These factors include the field of view (FOV) in degrees, the elevation of the station above sea level (CS_Z), and the resulting pixel resolution at distances of 150 and 450 meters for both the cross-shore

component (Ac) and the long-shore component (Ar).

| CS | focal (pix) | pixel size (ppp) | Target image width (pix) | FOV (º) | CS_Z (m) | Ac_150 (m) | Ar_150 (m) | Ac_450 (m) | Ar_450 (m) |
|---|---|---|---|---|---|---|---|---|---|
| *agrelo* | 6285 | 72 | 8320 | 67.00 | 7.1 | 0.02 | 0.44 | 0.06 | 3.98 |
| *cies* | 3200 | 72 | 4160 | 66.05 | 9.63 | 0.04 | 0.65 | 0.12 | 5.82 |
| *cadiz* | 1540 | 96 | 2048 | 67.24 | 16.96 | 0.09 | 0.76 | 0.26 | 6.84 |
| *samarador* | 2920 | 72 | 4000 | 68.82 | 3.06 | 0.04 | 2.20 | 0.13 | 19.84 |
| *arenaldentem* | 3000 | 72 | 4000 | 67.38 | 2.75 | 0.04 | 2.41 | 0.13 | 21.67 |

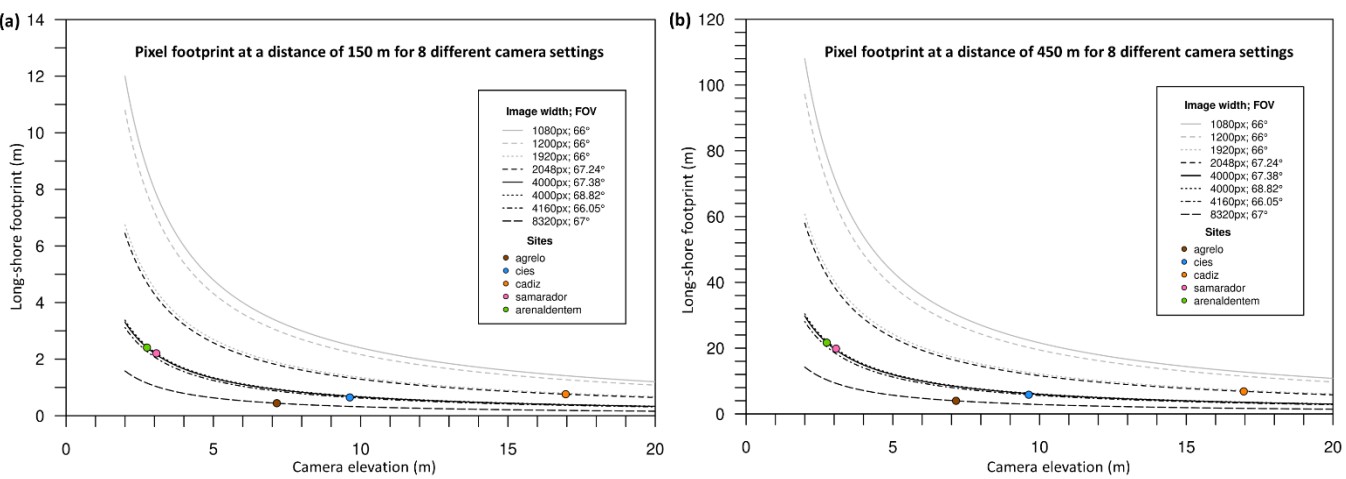

**Figure 5:** Long-shore footprint at distances of 150 m (a) and 450 m (b) for eight different camera scenarios, taking into account the elevation of the stations. The camera settings correspond to the typical social network images (grey) and the characteristics of the different target

images used in the present study (black). The use cases corresponding to the values of Ar in Table 3 are also represented (CS sites).

In the best scenario, the resolution of the long-shore pixel exhibits oscillations of approximately 0.44 m and 3.98 m at 150 m
and 450 m distance, respectively (*agrelo*, in Table 3). However, this resolution worsens to 2.41 m (at 150 m distance) and
21.67 m (at 450 m distance) in the case of lowest elevation *arenaldentem* CS (Table 3). Due to the increasing pixel footprint,

the accuracy of shoreline mapping is compromised, and manual digitization of the shoreline through visual photointerpretation
in the study sites becomes impractical beyond 450 m threshold. Consequently, data points beyond this distance were trimmed
and excluded from the analysis. Furthermore, the presence of site-specific geomorphological features, such as a greater slope
or curved planform, along with methodological constraints including the number and image distribution of the GCPs, as well
as the view of the beach from the station can contribute to increased errors during rectification and digitization processes.

Hence, it is imperative to adjust the shoreline cutoff distance based on the CS characteristics at each site to account for these
factors.

To enhance the overall precision of the extracted shorelines, a comparison was conducted between the water lines measured
using GPS RTK-GNSS and those derived from CS data (procedure named "first shoreline check" in data flow of Fig. 2). This
comparison involved analysing the XY deviation between the two datasets, considering both timestamp and site information.

The XYZ distance to the station, as depicted in Fig. 6, was considered during the assessment.

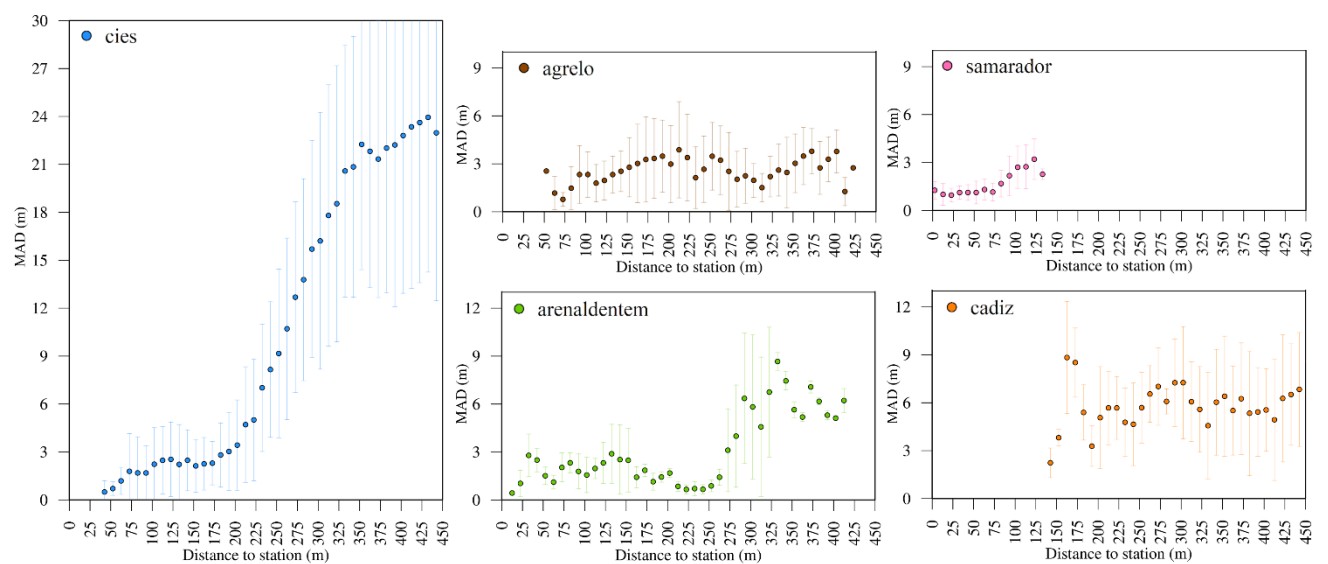

**Figure 6:** Mean absolute distance (XY) between CoastSnap and GPS RTK-GNSS coincident shorelines, at different sections along the distance to station (XYZ) and for each site within a cutting point range of [0, 450] m. Standard deviation is represented by the bars.


The comprehensive analyses conducted in this study contributed to the determination of the optimal XYZ distance between
shoreline points and the CS for precise and reliable shoreline mapping. In order to ensure the optimal representation of each





site, the extracted shorelines were carefully trimmed to the appropriate distances, as summarized and explained in Table 4. The influence of this cutoff distance on the retained shoreline length is shown in Fig. 7.


**Table 4:** Maximum distance to ensure shoreline mapping accuracy for each CoastSnap station in SCShores. See Fig. 7 for reference.

| CS | Maximum distance (m) | Additional considerations |
|---|---|---|
| *agrelo* | 360 | Visually covers until the northernmost river outflow. Highly affected by the beach face slope. |
| *cies* | 175 | Visually covers until the start of the dune sector. Highly affected by beach curvature, beach face slope, and the distribution of the available GCPs. |
| *cadiz* | 450 | Visually covers until the second curvature of the beach, at 450 m. Curvature and lower resolution impedes further accurate digitization. |
| *samarador* | 150 | Covers most of the beach shore; low station elevation and beach curvature limit further digitization. |
| *arenaldentem* | 275 | Highly affected by pixel resolution and shadowing effect of waves induced by view perspective and low station elevation. |

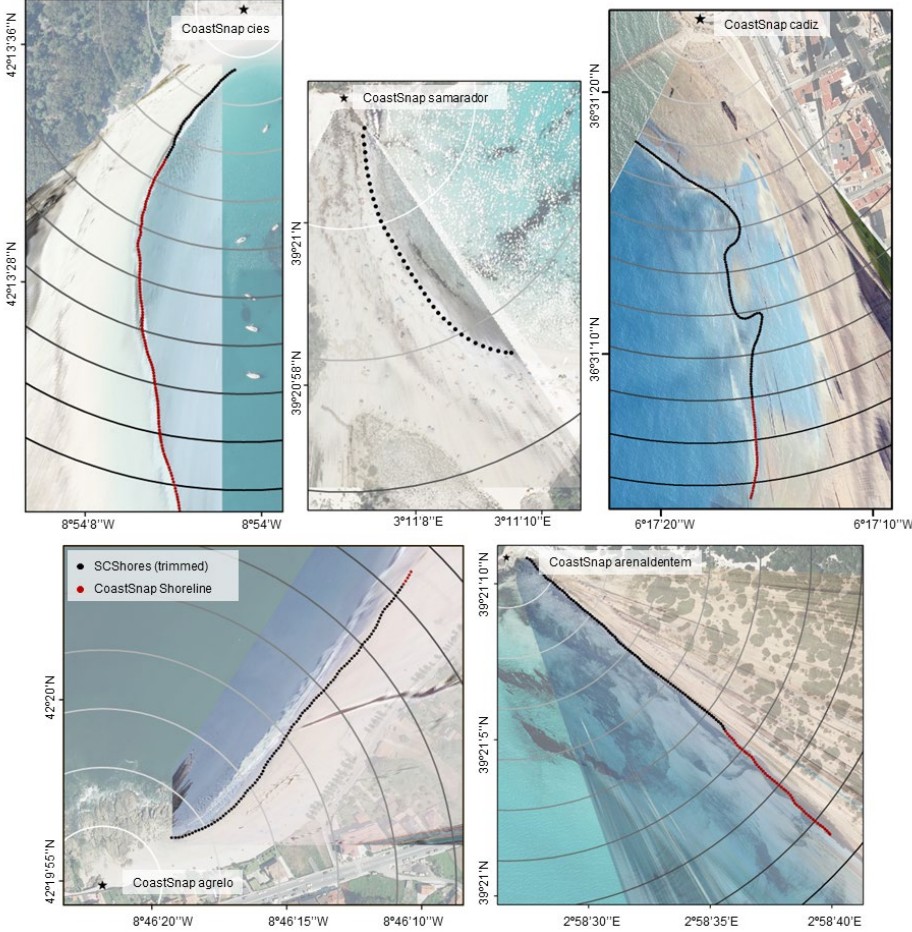



**Figure 7:** Rectified CoastSnap photographs for each site, showcasing the entire mapped shoreline (highlighted in red) and the resulting
shorelines after trimming (depicted in black) in accordance with the optimized distances detailed in Table 4. The isolines, delineated at 50-
meter intervals, denote the plain distances from the CoastSnap stations, thereby providing a spatial reference within the SCShores dataset.
Background for the rectified photos are orthophotos from PNOA sources (CC-BY 4.0 scne.es 2021). The grid coordinates are
GCS_WGS_1984.

The next step entailed homogenization of the coordinate reference system. All the shorelines were reprojected to a single
coordinate system, WGS84 (EPSG: 4326), as the data covered different UTM zones. Finally, all the shorelines were compiled
into a single dataset (SCShores v1.0) using the GeoJSON format (Butler et al., 2016), an open standard for representing
geographical features and their associated non-spatial attributes.

**2.2.5 Dataset validation**

For the final dataset validation, the same in-situ GPS RTK-GNSS shoreline measurements at CoastSnap sites used in the
previous section were utilized. These measurements adhered to the same criteria employed for mapping shorelines, which
involved identifying the dynamic shoreline. The validation process comprised several consecutive steps. Initially, a match-up
exercise was conducted to select the CoastSnap derived shorelines (SCShores_v1.0) that closely corresponded in time with the
in-situ measured ones acquired on the same date. The identified shoreline matches can be observed in Fig. 8. The average time
difference between the measurements was ±11.4 minutes. The number of match-ups per site was variable, with 2 GPS
measured shorelines for *arenaldentem*, 5 for *samarador*, 14 for *cies*, 11 for *agrelo*, and 3 for *cadiz*. To assess the accuracy,
the shorelines extracted from CoastSnap were trimmed to align with their corresponding GPS measured ones. This alignment
was achieved by defining a linear area that covered the maximum lengths of the measured lines. The trimmed shorelines were
then validated through a point-to-point analysis. For each point along a CoastSnap-derived shoreline, the minimum Euclidean
distance to the corresponding measured shoreline points was calculated as in Eq. (1):

$$\varepsilon = M_i - O_i , i = 1, ..., n \tag{1}$$

where $M_i$ are the SCShores mapped points and $O_i$ the measured ones, and $n$ is the number of points of the mapped shoreline.
Then, the $\varepsilon_i$ distances were used to calculate different evaluation metrics both globally and per site. These metrics include the
Mean Absolute Distance (MAD, Eq. 2); Root Mean Square Distance (RMSD, Eq. 3), the percentage probability of $\varepsilon$ being
less than or equal to 3 m (P3, Eq. 4), and the value corresponding to the 75th percentile of the computed distances (Q3).

$$MAD = \frac{1}{n}\sum_{i=1}^{n}|\varepsilon| \tag{2}$$

$$RMSD = \sqrt{\sum_{i=1}^{n}\frac{\varepsilon^2}{n}}, \tag{3}$$

$$P3 = \frac{n_{[|\varepsilon|\leq 3]}}{n} \times 100 \tag{4}$$



## 2.3 Dataset description

The SCShores v1.0 dataset is available in the form of a single geospatial layer, specifically a GeoJSON file. In total, the dataset comprises 1721 shorelines, which monitor a coastal extension of roughly 1.3 km between April 2018 and December 2022 (Fig. 8). The number of shorelines per site is variable, as outlined in Table 5, ranging from 40 for the site *arenaldentem* (active since July 2022) to 950 for the site *cadiz* (active since September 2020). The ~ 80 % of the acquired shorelines correspond to a TSQ rating of 1 (Table 5) signifying the high reliability of the obtained shorelines.

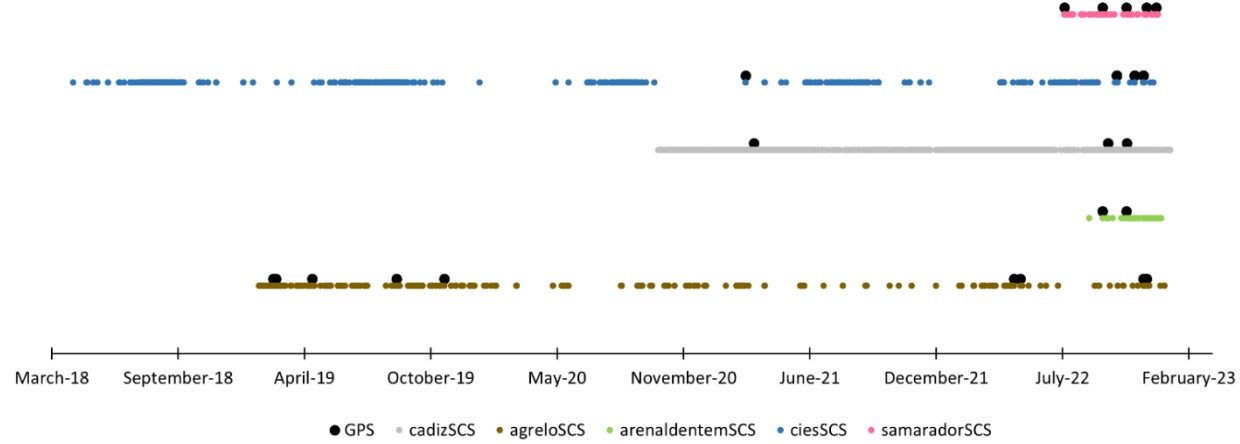

**Figure 8:** Timeline of SCShores_v1.0 coastlines and GPS RTK-GNSS measurements.

**Table 5:** Summary of SCShores_v1.0 content per study site. CS: name of the CoastSnap station; Nshores: number of shorelines per site in the dataset; meanNpoints: mean number of shoreline points per site; meanLength: mean length of the obtained shorelines per site in the dataset. TSQ1 and TSQ2 correspond to the percentage of shorelines with a timestamp flag equal to 1 and 2 respectively.

| CS | Nshores | meanNpoints | meanLength (m) | TSQ1 (%) | TSQ2 (%) |
|---|---|---|---|---|---|
| *agrelo* | 244 | 106 | 315.45 | 90.16 | 9.84 |
| *cies* | 430 | 42 | 124.04 | 53.72 | 46.28 |
| *cadiz* | 950 | 128 | 383.36 | 91.37 | 8.63 |
| *samarador* | 57 | 48 | 140.50 | 75.44 | 24.56 |
| *arenaldentem* | 40 | 86 | 255.51 | 85.00 | 15.00 |



The GeoJSON file consists of a feature collection of multipoint geometries (Lon, Lat, Z) in the WGS84 coordinate reference system (EPSG: 4326), corresponding to the mapped shorelines for each available date and time. Each shoreline in the collection
is associated with seven additional attributes (Table 6).

**Table 6:** Description of data fields for the SCShores_v1.0 dataset at individual shoreline scale.

| Attribute | Values | Description |
|---|---|---|
| *site* | e.g. agrelo, samarador, cadiz, …. | CoastSnap station name id (CS). |
| *date* | yyyyy-mm-dd hh:mm:ss | Date and time. |
| *timezone* | UTC | Universal Time Coordinated. |
| *timestampQuality* | 1, 2 | Quality flag providing the confidence in the date-time indicated by the image provider. |
| *imageSource* | Instagram, Twitter, Facebook, Email, CoastSnapApp | Source of the original image from which the shoreline has been derived. |
| *elevation_m* | Tide+tide offset | Same as Z coordinate, defined by the observed tide plus the tidal offset, in meters. |
| *verticalDatum* | NMMA | Mean sea level in Alicante, which is considered the vertical datum in the Spanish territory. |
| *geometry* | MultiPoint | Type of geometry used in the file. |
| *coordinates* | longitude, latitude, Z | Geographic WGS84 coordinates for each point in the geometry. |

## 3 Results and discussion

### 3.1 SCShores v1.0 performance assessment

Overall, the methodology used enabled the extraction of CS shorelines with approximately 67% probability of error being less than or equal to 3 m, and 75% being less than 3.8 m (Table 7). The Mean Absolute Distance (MAD) at different sites relative to the distance from each station and taking into account the varying CS elevations, is shown in Fig. 6.

**Table 7:** Descriptive statistics for individual sites and combined data. $N_{GPSshores}$ corresponds to the number of available in-situ GPS measured
shorelines per site and $N_{points}$ indicates the number of CS shoreline points compared to compute the statistics (see Sect.2.2.5).

| CS | $N_{GPSshores}$ | $N_{points}$ | MAD (m) | RMSD (m) | Q3 (m) | P3 (%) |
|---|---|---|---|---|---|---|
| *agrelo* | 11 | 1048 | 2.59 | 3.30 | 3.59 | 67.56 |
| *cadiz* | 3 | 335 | 5.95 | 6.68 | 8.5 | 20.00 |
| *cies* | 14 | 591 | 1.98 | 2.79 | 2.50 | 82.23 |
| *samarador* | 5 | 220 | 1.71 | 1.15 | 2.07 | 84.55 |
| *arenaldentem* | 2 | 175 | 1.68 | 1.14 | 2.20 | 86.29 |
| All sites | 36 | 2369 | 2.76 | 3.71 | 3.78 | 67.45 |



This study identifies several factors that contribute to the potential errors in shoreline extraction using the CoastSnap methodology. These factors include registration and image resolution, spatial resection process and rectification (involving the number and suitability of GCPs), tide bias, beach shape and morphology (e.g., planform curvature, beach-face slope or the presence of sand bars and troughs), and the position of the CoastSnap stations (Table 4); including the distance and orientation to the shoreline points.

The highest accuracies were achieved for microtidal sites, while the poorest results were obtained for the *cadiz* site. However, it is important to note that this does not necessarily imply that shoreline accuracy is inherently lower in mesotidal areas. The uncertainty associated with the swash zone in areas with a very low slope and the presence of bars such as Santa María beach (CS: *cadiz*), creates greater uncertainty when measuring (in field) and mapping the water line due to the presence of a water sheet. Therefore, the observed differences in accuracy are attributed to these specific challenges encountered in mesotidal zones rather than a limitation inherent to the CoastSnap methodology.

## 3.2 Limitations of the dataset

The limitations of the dataset relied on temporal and spatial inconsistencies, as well as uncertainties in the resulting shorelines already discussed in section 2.2.4. In this section we present the observed temporal inconsistencies between different CS which can be attributed to the varying installation times, as shown in Fig. 8, and citizen participation. The first station (*cies*) was installed in April 2018, while the last one was installed in September 2022, resulting in a difference in record start times of 52 months.

The dataset contains several gaps in certain stations that have different underlying causes. As the source of the images used to extract shorelines is citizen volunteers, any limitations, or restrictions to their access to the CS results in a cessation of the input flow of raw data. In 2020, the COVID-19 pandemic caused a three-month gap in the data collection, coinciding with the period of lockdown and mobility restrictions in Spain (Fig. 8). Additionally, there was a decrease in the number of images during the following months due to limitations on displacements. One particular case concerns the station located on the Cies Islands. These islands are uninhabited for most of the year, with a significant influx of tourists only during the summer season. As a result, the raw data is primarily concentrated during these months. Additionally, active engagement of the local community is a crucial factor in ensuring a steady flow of raw data inputs. This is highlighted when comparing the data from *cadiz* with the data from *cies* or *agrelo*, which have longer time records but less participation (Fig. 8). To increase citizen participation, the "Centinelas de la Costa" project has developed various fruitful strategies to engage citizens in the CoastSnap initiative. Finally, it must be taken into account that all the results derived from CoastSnap initiative, have to be fed into social media so that citizens can see the results of their collaboration, thereby encouraging participation.

The reliability of shorelines is dependent on the quality of the timestamp data (TSQ) associated with them, as shown in Table 6. TSQ value of 2 was primarily assigned to images sourced from social media platforms (such as Instagram, Twitter, and





Facebook) that are usually uploaded at the time of capture. However, these images are accompanied by a higher degree of
uncertainty than those that include a date and time (TSQ=1).

## 4 SCShores: Potential applications

High-resolution data with frequent time intervals is crucial to gain a comprehensive understanding of beach dynamics, leading
to effective management strategies and facilitating the application of future models. This dataset provides valuable information
on shoreline changes, enabling local authorities and beach managers to monitor erosion and accretion rates and implement
coastal protection measures. Furthermore, this dataset can be utilized to validate and fine-tune numerical coastal evolution
models. Lastly, this dataset can be a valuable resource for studying beach morphodynamics and the underlying forces that
shape it, such as tides and waves. This can help researchers better understand how physical processes influence beach dynamics
across space and time and further develop conceptual models. Some examples of SCShores dataset applications are provided
in Fig 9.
Figure 9a presents a proxy for the beach carrying capacity at the mesotidal beach of Santa María in Cádiz (*cadiz* CS), based
on its widths at different tidal stages. By analysing the area between the shoreline and the promenade, the dataset enables the
calculation of available beach space. Figure 9a left shows the linear regression analysis of the calculated area for each date
with respect to the estimated tide. It is important to note that the area is calculated with a fixed reference baseline, but the
shorelines do not cover the entire beach as Table 4 indicated. Therefore, the displayed area represents approximately 75% of
the alongshore extent of the beach. By means of the relationship (equation) derived from this analysis, Figure 9a right presents
the number of potential beach users, under the assumption of no beach bars, tidal terraces or other limitations. The calculation
depends on the "optimal conditions for bathers" or conditions imposed by measures like those related to COVID, across
predefined occupancy degrees of 5 to 25 $m^2$ per user in order to determine the maximum number of users (Murillo et al., 2023;
Zacarias et al., 2011; Zielinski and Botero, 2020). It should be noted that these data can be used as input in more sophisticated
studies on beach carrying capacity. By incorporating the sea level information measured by the tide gauge at any given moment,
coastal managers can accurately estimate the approximate beach area and determine the maximum number of users at sites
such as Cadiz, where significant variations are unlikely. This valuable insight will contribute to an effective management and
planning, optimizing the use of the beach resources while maintaining a safe and enjoyable environment for visitors.
Another application of the SCShores dataset is the estimation of beachface slope (Fig. 9b), calculated as in Eq. 6. The beachface
slope (steepness) is a fundamental parameter for coastal morphodynamic research as it is related to different morphological
characteristics and processes such as sand grain size, wave-run up elevation and total swash excursion at the shoreline (Vos et
al., 2020). In applications of this nature, it is crucial to take into account the provided performance metrics as a function of
distance, as illustrated in Fig. 6, as the error in ΔXY will significantly impact the calculated slope. The case of Rodas beach



(*cies* CS) presents the most complex or limiting scenario. It is worth noting the notable contrast in scale between the *cadiz* and *cies* CS sites (Fig. 9b). The former represents a typical dissipative beach, while the latter corresponds to a reflective beach.

$$\beta = \frac{\Delta Z}{\Delta XY} \qquad (6)$$

where $\Delta Z$ represents the elevation difference between two CS shorelines on the same day (low and hight tide); and $\Delta XY$ represents the difference in cross-shore position between these two CS shorelines. It is assumed that the shoreline elevation remains constant along the entire shoreline when calculated using CoastSnap.

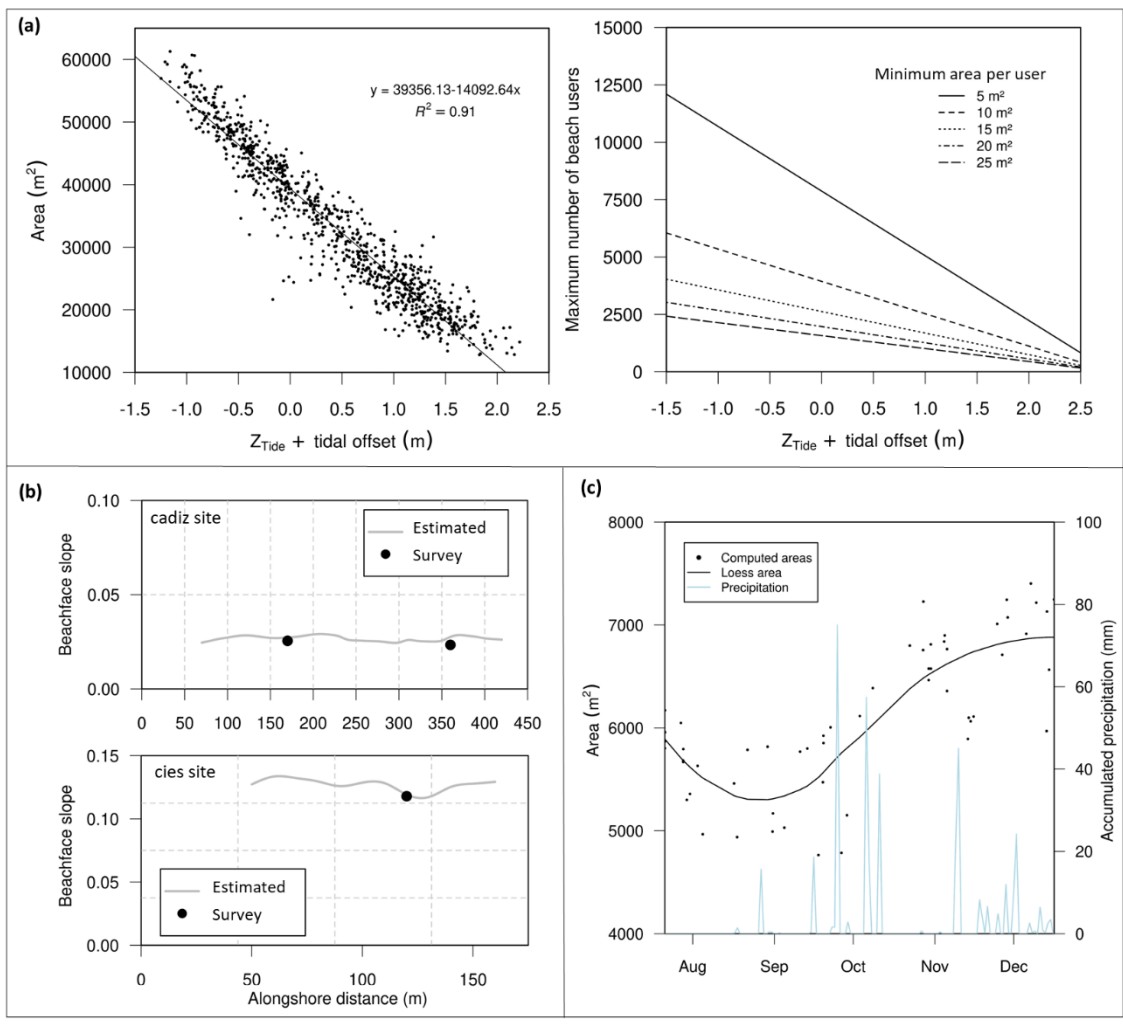

**Figure 9:** (a) Relationship between the tide and the maximum number of beach users at Santa Maria beach (*cadiz*), considering various scenarios of minimum area per user (see text for explanation). (b) Average beachface slope for the *cadiz* and *cies* sites, obtained from SCShores (estimated values) and measured in situ (survey). The slope derived from GPS RTK-GNSS measured profiles was computed by considering the cross-shore distance between the profiles' points at the same Z as the estimated elevation of CS shorelines for each date. (c) Comparison of the computed area using CS shorelines from *samarador* and the monthly accumulated precipitation data from the closest AEMET meteorological station.

The last example shows the evolution of the area of S'Amarador beach computed for all available shorelines (Fig. 9c). As S'Amarador beach is a microtidal beach (recorded sea level elevations ranging from 0.07 cm to 0.37 cm in the dataset) and the

beach landward limit for the computation of the beach area has been defined by the position of the dune foot, the area of the beach is modulated by the shoreline displacement. In this way, assuming negligible tidal effects, beach area can be considered a proxy for beach width. For mesotidal beaches, beach width can also be calculated, but it is necessary to consider the tidal range. This can be done either by considering similar tidal ranges or by referencing all shorelines to a common Z datum (Almonacid-Caballer et al., 2016) using the slope of the beachface, which can also be retrieved with the provided dataset as

shown in prior applications (Fig. 9b).

The evolution of S'Amarador beach area with overlapped records of monthly accumulated precipitation (Fig. 9c) shows that significant rainfall events may favour beach accretion. The beach is linked to a ravine that can become active during torrential rainfall events, transporting sediment to the beach if the intensity and duration of the event are sufficient. This analysis contributes to the geomorphological knowledge of beach behaviour and demonstrates that SCShores dataset can be

complemented with other forcing drivers to understand the key parameters controlling changes in coastal evolution.

## 5 SCShores: Usage recommendations

The dataset is presented in a user-friendly format to ease its importing into widely-used geographical information system (GIS) softwares such as QGIS and ArcGIS. In QGIS, the GeoJSON file can be added to the Layers Panel either by dropping it from the Browser Panel or by double-clicking on the file. However, in ArcGIS environment, the Data Interoperability extension is

required and must be activated prior to use.

By importing the dataset into GIS software, end-users can take advantage of the powerful functionalities offered by these platforms to visualize, manipulate, and extract valuable insights from the shoreline data. GIS software provides tools for spatial analysis, data integration, and visualization, enabling users to examine the dataset in conjunction with other relevant geospatial information.

Additionally, the dataset can also be accessed programmatically (e.g., Python, R), allowing for seamless integration into custom software applications or research workflows. This programmable accessibility enables coastal researchers to incorporate the dataset into their own analysis pipelines, automate data processing tasks, and facilitate the development of specialized tools and models.



**6 Code availability**

The source code to rectify and map shoreline derived from crowdsourced smartphone images (CoastSnap) is available at https://github.com/Coastal-Imaging-Research-Network/CoastSnap-Toolbox and technical methodology is described in Harley et al. (2019).

**7 Data availability and upgrade**

The SCShores v.1.0 dataset (González-Villanueva et al., 2023b), described in Sect. 2.3 of this article, is accessible via the
Zenodo data repository at https://doi.org/10.5281/zenodo.8056415 This dataset serves as a starting point for storing future shorelines gathered through the CoastSnap initiative in Spanish territories and for disseminating them to the scientific community and end-users. The authors will periodically update the database as new data from in-situ shorelines measured with GPS RTK-GNSS becomes available for the validation process. New versions of SCShores will be uploaded to Zenodo, utilizing the repository's versioning feature.

**8 Conclusions**

This study presents a novel dataset of shorelines derived from a citizen science initiative known as CoastSnap, focusing on five Spanish beaches spanning both meso- and microtidal environments. The dataset encompasses a comprehensive collection of 1721 shorelines, each represented by a multipoint feature with a regular spacing of 3 m. Notably, an assessment of the data quality reveals a compelling accuracy profile, with an estimated probability indicating that approximately 67% of the
measurements exhibit an error magnitude within 3 m, while 75% demonstrate an error magnitude within 3.8 m.

The presented dataset provides invaluable insights into the local-scale variability of shoreline positions, serving as a unique and cost-effective resource for beach monitoring programs, especially in regions where in-situ observations are limited or unavailable. This dataset holds significant potential for various applications in coastal science and management, including:

    i.    Estimation of variability in beach width and area: The dataset allows for the estimation of beach width and area
460          variations over time. This information is crucial for assessing beach carrying capacity and understanding beach evolution patterns.

    ii.   Enhanced understanding of short to medium-term coastal evolution: By analysing the dataset, researchers can gain a deeper understanding of the short-term coastal evolution processes and their relationships with driving forces such as wave action, sediment transport, and shoreline dynamics. This knowledge contributes to improved predictive models
465          and more accurate coastal management strategies.

iii.     Estimation of beachface slope: The dataset enables the estimation of beachface slope, a critical parameter required for multiple models used in coastal evolution studies. Accurate measurements of beachface slope contribute to better modeling of sediment transport, beach nourishment projects, and erosion forecasting.

The utilization of this dataset in the above applications enhances the understanding of coastal dynamics and facilitates informed decision-making in coastal planning, and conservation efforts. It presents a valuable tool for researchers, coastal engineers, and policymakers to address the challenges associated with coastal evolution and sustainable coastal zone management.

The dataset is conveniently available in a user-friendly format, making it easily importable into commonly used geographical information system (GIS) software. By providing data accessibility through both GIS software and programmable interfaces, the dataset ensures that coastal researchers and end-users have the flexibility and convenience to exploit the data in a manner

that best suits their specific requirements.

**Author contributions.**

RGV, JSG and ESG devised the study, designed the figures, and wrote the paper. IA, FCS, TP, AFM, JB, LDR and MAN provided in-situ measurements. RGV, ESG and JSG processed the raw data. RGV and ESG mapped the shorelines and JSG compiled the dataset. All authors discussed the results and reviewed the paper.


**Competing interests.**

The contact author has declared that neither they nor their co-authors have any competing interests.

**Acknowledgements.**

Tidal data have been supplied by Puertos del Estado (Spanish Government). Orthophotographs have been supplied by the Instituto Geográfico Nacional (IGN) of Spain. The authors are grateful for the collaboration and support of the local Government of Bueu and the Ministry of Environment and Territory of the Balearic Islands, as well as the funding provided by the European Funds for Balearic Islands, University, and Culture. They also extend their gratitude to the Es Trenc-Salobrar de Campos Maritime-Terrestrial Natural Park and Mondragó Natural Park, and the Atlantic Island Galicia Maritime-Terrestial

National Park along with their dedicated technical staff.

This work is a contribution to IGCP Project 725 'Forecasting Coastal Change'; CRUNCH project (FEDER-UCA18-107062), funded by EU (2014-2020 ERDF Operational Programme) and by the Department of Economic Transformation, Industry, Knowledge, and Universities of the Regional Government of Andalusia; CRISIS project (PID2019-109143RB-I00), funded by the Spanish Ministry of Science and Innovation and the EU; and to Andalusian PAI research group RNM-328.

This work received the support from the Spanish Foundation for Science and Technology (FECYT) through the project "Centinelas de la Costa" (FCT-20-15835). This work has received financial support from the Xunta de Galicia (Centro de





Investigación de Galicia accreditation 2019-2022) and the European Union (European Regional Development Fund - ERDF). The installation of Cadiz station was funded by University of Cadiz through project PR2019-022.

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
