# Peer review of "SCShores: A comprehensive shoreline dataset of Spanish sandy beaches from a citizen-science monitoring program"

_Earth System Science Data, 2023_

## Author Comment (AC1)

**Reply to comment on essd-2023-230 (**https://doi.org/10.5194/essd-2023-230-RC1**) by an Anonymous Reviewer.**

We extend our appreciation to the referee for the constructive feedback and insights pertaining to our research. We provide a point-by-point response (R) to the reviewers' questions (Q). :

**Comment 1:**
Q: Would you clarify the rectification process (transform the image from pixel coordinates u, v to world metric coordinates X,Y,Z) in the data processing section.

R: Thank you for your remarks. The principal aim of this manuscript was not to provide an exhaustive elucidation of the methodological intricacies of CoastSnap. Instead, emphasis was placed upon the practical application of the methodology. The rectification process, encompassing the conversion from the image coordinate system to the terrain coordinate system, is a firmly established mathematical procedure. This procedure was previously elucidated by Harley et al. in 2019 (https://doi.org/10.1016/j.coastaleng.2019.04.003) within "Section 2.2.1: Image Georectification." The aforementioned article was intended to set the basis and the comprehensive methodological framework of CoastSnap in its entirety. The inclusion of the methodology would unnecessarily lengthen the manuscript, as it would constitute a repetition of the article previously published by Harley et al. 2019. In the submitted version of our manuscript this procedure was summarized and the reference of the original work was added, specifically within lines 199-204 (Section 2.2.3 Data processing).

*Reference: Harley, M. D., Kinsela, M. A., Sánchez-García, E., and Vos, K.: Shoreline change mapping using crowd-sourced smartphone images, Coast. Eng., 150, 175–189, https://doi.org/10.1016/j.coastaleng.2019.04.003, 2019.*

**Comment 2:**
Q: The reviewer recommends calibrating your model with high-resolution temporal satellite images for at least two of the studied five coasts (one for the meso-tidal coasts and one for the micro-tidal coasts). It is favorable to do this calibration for the Cadiz site due to shortage of the data during most of the year.

R: We agree with the reviewer, but it is of notable significance to underscore that the calibration procedure has been meticulously contemplated to be executed through the utilization of GPS in-situ data, primarily owing to its discernibly elevated precision levels. It is imperative to underscore that the employment of satellite imagery for the purpose of coastline detection at sub-pixel level necessitates a preliminary calibration procedure, effectively leveraging in situ GPS data. Furthermore, it is imperative to acknowledge that engagement with satellite imagery characterized by exceedingly high resolutions would invariably entail a transition toward proprietary entities offering data for monetary compensation, a trajectory that significantly diverges from the fundamental objectives intrinsic to the CoastSnap open citizen science initiative.

**Comment 3:**
Q: There are several factors affecting the coastal shorelines should be take in consideration within the produced model (especially in the micro-tide shorelines) e.g. waves, winds, thunder rainfalls etc....

R: The referee's viewpoint aligns with our perspective. It is crucial to know the dynamic interplay of factors, including wave dynamics, wind effects, and related variables, which significantly

shape the shoreline's positioning. Moreover, it is essential to recognize that the acquisition of such data mandates the establishment of measurement networks at each geographical location, complete with specialized instrumentation or modelling and the requisite resources for maintenance and data analysis. It is imperative, however, to note that these measurement networks are not universally accessible across all monitoring scenarios.

The methodology used within this manuscript seeks to uphold cost-effectiveness and underscore its applicability in regions where direct access to specific in-situ beach monitoring data or knowledge in modelling software is limited or where financial constraints prevail. Consequently, within the confines of this study, a localized tidal offset unique to each CoastSnap station is implemented. This approach serves to refine elevation approximations for the shoreline, thereby advancing the precision of its XY coordinates. Such strategic deployment of this methodology remains steadfast, even in light of the inherent intricacies inherent to the underlying phenomena.

We have introduced several lines of clarification within Section 2.2.1, titled "Data collection" to expound upon the assumed corrections applied during the shoreline extraction process. The modifications to the text have been incorporated within lines 170 to 176 of the document, as follows:

*"It is important to note that precisely resolving the complex interplay among various influential factors in shoreline elevation (e.g., bathymetry, significant wave height, wave period, wave direction, winds, run-up) requires the utilization of sophisticated models that rely on in-situ data. However, such data is often deficient, as is evident in the study areas under consideration. While this circumstance might introduce a degree of uncertainty, the computation of the vertical tidal offset employed in generating the provided dataset is inherently straightforward. This approach aligns with the corrective methodology proposed within the CoastSnap framework, and mitigates errors associated with the aforementioned factors in the estimation of shoreline elevation. (Harley et al., 2019)."*

**Comment 4:**
Q::With respect to the figures 1, 4 and 7; the reviewer recommends to enlarge the images in order to clarify its content.

R: Thank you for the comment, but the authors do not fully comprehend the meaning of "enlarge the images to clarify content." The images were created in accordance with the journal's specifications and submission guidelines. We have diligently prepared all figures at a resolution exceeding 300 dpi, with the intention of subsequently submitting them as individual files for the production phase, thus ensuring their superior quality. This measure is explicitly aimed at enabling potential enlargement, as per the specifications outlined by the ESSD journal, which dictates that figures should not exceed a width of 8 cm.

Nonetheless, new versions of the figures have been generated in an attempt to magnify the graphical content while seeking enhancement. We have included the versions that have been created for your reference, so that the reviewer can indicate which of these improvements aligns more closely with their request.

Options for Figure 1:

Version 1:

[Figure]

Version 2:

[Figure]

Figure 4:

[Figure]

**(a) CoastSnap cadiz**   Registered image          Rectified image

30 Nov 2022
*Falling tide leaving beach cusp formations on the wet sand*

100 m

11 March 2021
*Low tide with flooded areas (inter-tidal terraces)*

100 m

**(b) CoastSnap arenaldentem**   Registered image          Rectified image

27 Oct 2022
Calm swell

50 m

24 Nov 2022
Posidonia berms formation

50 m

Figure 7:

[Figure]

---

## Author Comment (AC2)

**Reply to comment on essd-2023-230 (**https://doi.org/10.5194/essd-2023-230-RC2**) by an Anonymous Reviewer.**

We extend our appreciation to the referee for the constructive feedback and insights pertaining to our research. We provide a point-by-point response (R) to the reviewers' questions (Q). :

**Comment 1:**
Q: In the description section of the monitoring stations, can you report if the support for the phone is anti-vibration (in case of wind)? If not, are the images acquired with wind then eliminated from the dataset?

R: The cradles for the bases have been constructed using 6mm grade 316 stainless steel, known for its exceptional durability. These cradles have been securely anchored to pre-existing infrastructures (Cies, Agrelo, and Cadiz), ensuring their fixed maintenance. For the remaining two bases in the Balearic Islands (Arenaldentem and Samarador), wooden posts measuring 1.10-1.20 meters in height were installed. These posts are firmly attached to metal anchors, which in turn are secured to the rocky substrate using galvanized bolts. This robust setup guarantees the stability of the bases, even under conditions of strong winds. Nonetheless, all locations undergo periodic reviews to execute essential maintenance tasks and preserve the installations' initial condition.
Additionally, during the registration process of the images to a fixed target image, any minor deviations are corrected. As a result, the removal of images has not been necessary due to wind-induced vibrations. Instead, removals have been attributed to the utilization of zoom, filters, or incorrect positioning during photo acquisition by users.

To enhance clarity on the cradle designs, we have introduced several clarifications within Section 2.1 titled "Study sites and CoastSnap station settings." The text revisions have been inserted between lines 93 and 95 of the document, as outlined below:

"*A thickness of 6mm was selected for the grade 316 stainless steel used in crafting the CoastSnap bases. These bases have been securely affixed to existing infrastructures or dedicated wooden posts bolted to rocky substrates, ensuring their stability and preventing any movement.*"

**Comment 2:**
Q: In the section of the image processing. could you indicate whether these GCPs are located in "fixed" positions/locations(e.g. rocky outcrops, man-made structures, etc.)

R: Thank you for your input. We have included a brief explanation in the main text at the conclusion of the "2.1 Study Sites and CoastSnap Stations Settings" section, following the description of the number of points measured at each station. Now in lines 140 to 144 as follows:

"A*ll GCPs selected for this study correspond to fixed points that are easily distinguishable within the target image. These GCPs encompass various types of features, including natural elements such as rocks and outcrops, as well as man-made structures like houses and maritime or beach infrastructures. This careful selection of identifiable and well distributed GCPs contributes to the accuracy and reliability of image analysis processes.*"

**Comment 3:**

Q: Please, could you better describe the procedure for calculating the tidal offsets. In my opinion it is not clear how the correction takes place.

R: Thank you for your input. We have included additional sentences in the main text to clarify the offset calculation process. We have revised the text in the main document's 2.2.1 Data Collection section as follows, in lines 166 to 175:

Previous text:

*Due to the absence of local tide gauge at the beach, local tidal offsets were calculated for each beach by computing the difference between the in-situ measurements obtained using GPS RTK-GNSS and the corresponding tide gauge records for the same time. The validity of the obtained offsets was subsequently established by cross-checking these in-situ tidal elevation measurements against the corresponding tidal elevation value of the timely-closest available CoastSnap image (Fig. 3).*

New text:

"*Due to the absence of a local tide gauge at the beach, we determined local tidal offsets for each beach. This was accomplished by comparing the in-situ measurements acquired using GPS RTK-GNSS with the corresponding records from the tide gauge for the same time period. Tidal height measurements on the beach were obtained simultaneously while measuring the waterline using GPS RTK-GNSS, aiming to cover various meteorological scenarios, including both high and low-pressure systems, as well as varying wave conditions. The differences between the two datasets were calculated and averaged by site. The accuracy of the determined local offsets was later confirmed by comparing for the same dates, these shoreline elevation measurements ($Z_{GNNS}$) with their corresponding estimated tidal values ($Z_{Tide} + tidal offset$) used as the CoastSnap shoreline elevation (Fig. 3)*"

**Comment 4:**

Q: In the section of potential application, can the presented tool be also useful for measuring and characterize in morphology the banquette accumulations (Posidonia berms) along Mediterranean beaches? If the authors suppose that the presented tool are suitable for that, a brief description may be provided.

R: We appreciate your input. While the proposed application is certainly thought-provoking, it does not constitute the primary focus of this study. Our main objective is to present a comprehensive dataset of the coastline, accompanied by a thorough explanation of the procedural methodology employed in its creation and its potential uses or applications. Addressing the concern raised by the Reviewer would require not only the coastline dataset as provided, but also the related oblique images used to derive the shorelines. However, including these images in the dataset is not feasible due to privacy concerns involving beachgoers.

While we will take the Reviewer's suggestions into account for potential future research, it's important to emphasize that the current dataset's state does not allow for such an implementation. This possibility extends beyond the scope of the current data paper

---

## Author Response (AR2)

Dear Dr. Alberto Ribotti,

We have submitted the revised version of the manuscript titled 'SCShores: A Comprehensive Shoreline Dataset of Spanish Sandy Beaches from a Citizen-Science Monitoring Program' (essd-2023-230).

Below, we have provided a list of changes made based on your suggestions:

**Comment 2:** for completeness in decription, please add a few lines explaining why you cannot calibrate your model with high-resolution temporal satellite images

*We agree with the suggestion and have incorporated new text into section 2.2.4, titled 'Dataset Compilation,' to address the reviewer's comment. This new text can be found in lines 286 to 290 in the document with tracking changes.*
*"Validation solely utilized GPS RTK-GNSS measurements due to their widespread availability and unmatched precision (≤ 15 cm) across all three spatial dimensions (X, Y, Z). High-resolution satellite images (< 50 cm² pixel size), although accessible, require funding as they originate from commercial platforms like WorldView2/3. Their spatial accuracy relies on both pixel size and georeferencing precision, introducing potential uncertainties in shoreline position and the further validation of CoastSnap shorelines."*

**Comment 4:** the referee asked for the readability of certain figures to be improved, not the resolution, which is known to the authors when each part is created. The figures, once published, will be reduced in size impairing readability in some of their parts and thus losing some or all of their explanatory function. I therefore ask the authors to improve the readability of the following figures, even if improved with version 2, in particular (version 2):
- figure 4: the lettering inside the photos is illegible. The small map at the bottom left of the high left larger map is already illegible now. I also suggest to change the colors of the stations (now orange and light blue).
- figure 5: the legend in both figures should be increased in size and the lettering placed in bold. Lines ranging from 1080 to 1920 px are invisible even now
- figure 7: the co-ordinates along the axes of the figures are almost illegible, as are the inscriptions inside and of the legend, which therefore need to be revised as for the previous figures

*We agree with the suggestion and have improved the indicated figures. We have selected version 2 of Figure 1, as included in the Reviewer's response, and have implemented all the changes suggested by the Topic Editor. Additionally, we have made the necessary changes to Figures 4, 5, and 7. Furthermore, we have adjusted the text size in Figure 6 to enhance readability*

We have thoroughly revised the manuscript and considered the topic editor and reviewers' suggestions, which we have found very useful, and we hope this new version of the MS is worthy of publication in *ESSD* Please, if you have any further questions, do not hesitate to contact me at any time.

Sincerely,

Rita González-Villanueva, on behalf all the authors.

---

## Author Response (AR3)

Dear Dr. Alberto Ribotti,

We have submitted the final version of the manuscript titled 'SCShores: A Comprehensive Shoreline Dataset of Spanish Sandy Beaches from a Citizen-Science Monitoring Program' (essd-2023-230).

On behalf of all the authors, we would like to express our gratitude for the promptness and feedback received in regard to the current work. Additionally, we would like to extend our appreciation to the anonymous reviewers for their comments, suggestions, and the swiftness of the process.

Sincerely,

Rita González-Villanueva, on behalf all the authors.